# Optimum Bloating-Activation Zone of Artificial Lightweight Aggregate by Dynamic Parameters

**DOI:** 10.3390/ma12020267

**Published:** 2019-01-15

**Authors:** Young Min Wie, Ki Gang Lee

**Affiliations:** Department of Materials Engineering, Kyonggi University, Suwon 162-27, Korea; supreme98@kyonggi.ac.kr

**Keywords:** bloating clay, artificial lightweight aggregate, montmorillonite, rapid firing, normal sintering, optimum bloating-activation zone

## Abstract

The purpose of this study is to compare the bloating mechanism of artificial lightweight aggregate under sintering and rapid sintering conditions to identify the factors behind the bloating of the lightweight aggregate under these sintering conditions, and to find suitable temperature ramping conditions. The aggregate had an average particle size of 10 mm as formed using acid clay, and it was fired by a rapid sintering method and a normal sintering method. The bulk density and water absorption ratio of the specimen were measured, and the cross section was observed. No black core was observed under the rapid sintering condition, and it was lightened at an inflection point of 1150 °C. A reduction in the bulk density was observed in a shorter period of time when the input temperature was high under the normal sintering conditions. Regardless of the input temperature, the bulk density change was divided into three sections and a bloating-activation zone was observed in which the density abruptly decreased.

## 1. Introduction

Artificial lightweight aggregates are used in skyscrapers, super-long bridges and offshore structures. They are used as building materials for various purposes due to their high thermal insulation capabilities and light weights compared to those of ordinary aggregates. Artificial lightweight aggregate materials on the market are mainly produced by sintering in a rotary kiln using slate or shale. Recently, many studies have focused on various types of solid waste such as plastic waste [1], sewage sludge [2], coal-fired power plant ash [3,4], drill cuttings [5], mining tailings [6] or carbon fiber mineral waste [7].

For lightweight aggregate materials, the bloating mechanisms have been investigated in a range of studies to examine physicochemical characteristics and processes.

First, in terms of the physicochemical aspects, Riley [8] investigated various clays in the state of Minnesota in the United States (USA) and presented the chemical compositions of lightweight aggregates in a ternary phase diagram. Cougny [9] expressed bloating regions in the phase diagram of Al_2_O_3_-flux-Fe_2_O_3_, with fixed silica content to emphasize the role of Fe_2_O_3_ in Riley’s Diagram. Park et al. [10] studied the black core phenomenon caused by the reduction of Fe_2_O_3_ in an aggregate material and reported that the reduction of Fe_2_O_3_ in the aggregate and the generation of oxygen are the main causes of aggregate bloating. Cougny [9] also proposed particle size distribution of raw materials to make lightweight aggregate.

With regard to the processes, Riley [8] noted that most clay minerals cause foaming during rapid sintering. Lee [11] found that if the particle size of the pellet is relatively small, the degas rate becomes higher as the specific surface area increases, making it difficult to expand. In Kang et al. [12], a bloating mechanism that does not produce a black core under sintering conditions was disclosed, and was found to be possible to obtain more homogeneous pores although a high energy cost is incurred compared to a rapid sintering condition. Finally, Dondi [13] conducted a study which applied the Riley and Cougny diagrams to data in the literature on a lightweight aggregate made from waste materials. The chemical composition of the liquid phase, the high-temperature viscosity and the atmosphere of the furnace were more important than the total chemical composition of the raw materials.

In these previous studies, the proper bloating conditions of lightweight aggregate were noted. The proper chemical composition for bloating, the appropriate viscosity of the material at the bloating temperature, and the generation of gas should be assessed. The bloating mechanism stemming from the formation of the black core by the reduction reaction of Fe_2_O_3_ is accepted as orthodoxy. Regardless of the generation of the black core, rapid sintering is advantageous for trapping the gas that is degassed at the calcination temperature. However, in large rotary kilns in the field, it is difficult to create the conditions required for rapid sintering. Therefore, it becomes simply a normal sintering condition. It is necessary to clarify the bloating mechanism under sintering conditions and to confirm the bloating-activation zone. Here, the bloating-activation zone is defined as the time at which a rapid density reduction occurs when meeting the conditions for bloating under normal sintering conditions.

Therefore, in this study, we investigated the mechanism of bloating under normal sintering conditions and evaluated the suitability of acid clay materials. In order to find the optimum bloating-activation zone, changes in the physical properties of the aggregate were observed while changing the input temperature and the heating time, and an optimum bloating-activation zone of lightweight aggregate was established at the normal sintering condition.

## 2. Experimental Method

### 2.1. Raw Material Analysis

The acid clay used in this experiment was manufactured by Donghae Chemical Industrial Co., Ltd (Seoul, Korea). The chemical composition was analyzed by X-ray fluorescenceXRF (ZSR-100e, Rigaku, Tokyo, Japan) as shown in Table 1. An X-ray diffractometric (XRD, Miniflexll, Rigaku, Tokyo, Japan) analysis was performed to assess the crystal phase. After this analysis, the peak was found using the correct joint committee on powder diffraction standards (JCPDS) card to analyze the minerals. A thermogravimetry-differential thermal Analysis (TG-DTA, STA-409, Netzsch, Berlin, Germany) analysis was also conducted to investigate the weight reduction and gas generation reaction of the raw materials. Particle size analyze (PSA, Mastersizer 3000E, Malvern, Malvern, UK) analysis was done by laser diffraction to determine the particle size distribution of the raw materials. The liquid limit (LL) and plastic limit (PL) of the acid clay was measured by KSF2304 [14].

### 2.2. Aggregate Molding and Firing

For the firing test, 10 mm spherical aggregates were shaped by hand using acid clay. The molded aggregate was dried at 105 °C until the weight became constant. Thereafter, the firing test was carried out using a box-type electric furnace. There are two methods of firing; first, sintering was carried out by a rapid firing method. The rapid sintering method involves the use of an electric furnace, which is maintained at the maximum temperature, molded bodies are put in, and maintained for 10 min. The sintering temperature was in the range of 1100–1200 °C. Second, sintering was carried out by a normal sintering method. In order to simulate the firing conditions of a rotary kiln in the normal sintering experiment, the input temperatures were fixed at 300 °C and 600 °C, the temperature was raised to 1200 °C immediately and firing was then done but not then maintained at a maximum temperature. We then naturally cooled down the furnace. The sintering times ranged from 30 to 330 min at a starting temperature of 300 °C and from 30 to 270 min at a starting temperature of 600 °C. Table 2 summarizes the experimental conditions of the sintering test. The bulk density levels of the fired aggregates were measured according to KS F 2503 [15], where bulk density is the density in the oven-dry condition defined by the JIS A5002 [16]. The cross section of the measured sample was observed using an optical microscope.

The bloating index (BI) was calculated as the volume changes after heating (pre-firing + firing steps), according to the equation BI = 100 × (d_2_ × d_1_)/d_1_, where d_1_ and d_2_ are the diameters of the granules before and after heating, respectively [17,18].

The single aggregate crushing strength was measured by Universal Testing Machine (DS-001, Daeshin, Namyangju, Korea). The single aggregate crushing strength value (S) was determined according to the equation S = (2.8*P_c_*)/ (πX^2^), where *P_c_* is the load at which a rupture occurs and X is the sphere diameter [19,20]. The average single aggregate crushing strength was calculated from the tests performed on 15 granules.

## 3. Results and Discussion

### 3.1. Raw Material

A chemical analysis of the raw acid clay revealed the presence of Fe_2_O_3_ at 2.9 wt %. This is somewhat lacking in the titration results from the Cougny diagram (Figure 1). Park et al. [10] reported that a black core forms due to the reduction reaction of Fe_2_O_3_ and that FeO and unburned carbons are present in the black core. Therefore, for the acid clay, the expansion is not due to a black core but is instead due to another mechanism. The results of the XRD analysis of the samples are shown in Figure 2. A peak indicating montmorillonite appeared around 2°, 10° and 12°. The subsequent peaks were similar to albite. The high peak around 28° is considered the main peak of quartz. The major peaks for each crystal phase are shown in the figure. The major phases of acid clay used in this experiment are montmorillonite, albite and quartz.

The TG-DTA results (Figure 3) show that interlayer water evaporates from 100 to 200 °C; thereafter, the weight reduction due to the detachment of crystalline water continued until 1000 °C. The dehydration reaction of clay minerals differs depending on the type of raw materials used. In the case of halloysite, the interlayer water in the raw material evaporates between 100 and 300 °C, and the level of crystalline water intensively drops sharply at around 600 °C. In the case of montmorillonite, it is known that the level of crystal water drops slowly up to 1200 °C [21]. It is considered that montmorillonite is contained in the major minerals of the acid clay, and it readily bloats at a high temperature due to the continuous detachment of the crystal water at a high temperature given the characteristics of this mineral. Riley [8] concluded that minerals classified as the illite type are more advantageous for bloating due to the detachment of crystalline water at high temperatures. Illite minerals are pyrophyllite minerals, and dehydration of the crystalline water takes place slowly like acid clay used in this experiment.

A PSA analysis was conducted to determine the particle size distribution of the raw materials (Figure 4). As a result, it was found that most of the acid clay materials had a particle size of less than 50 μm. According to Cougny [9], more than 50% of the particles have a particle size of 1 μm, which is necessary for making lightweight aggregates. Nevertheless, the analysis showed that insufficient numbers of microparticles were detected. However, the study of Cougny [9] was carried out by the sedimentation method and should not be directly compared with the experimental data. As a result of the experiment, the molded body was well formed and the aggregate foamed well.

The Atterberg limits were measured, and shown in Table 3. The PL in clay is the minimum water content at which the plasticity of the clay appears, and the LL is the point at which it begins to flow like a liquid beyond the plastic zone. Moreno-Maroto et al. [22] classified clay by PI/LL. The PI/LL value of the acidic clay used in this experiment is 0.27, and it is classified as a low plasticity clay with a PI/LL value of 0.2 < x < 0.33. Although the acid clay used in this experiment can be molded if it contains a certain amount of water, it is judged that the plasticity is not relatively high.

### 3.2. Rapid Sintering

Generally, the bloating mechanism of lightweight aggregate is known to be the formation of a black core caused by the reduction of the inside of the aggregate [10]. Bloating due to the formation of black cores is known to reduce Fe_2_O_3_ at high temperatures to produce FeO, and the resulting gas increases internal pressure. The equation for this is as follows:
(1)$$3{\mathrm{Fe}}_{2}O_{3}\left(s\right)\to 2{\mathrm{Fe}}_{3}O_{4}\left(s\right)+\frac{1}{2}O_{2}\left(g\right)$$
(2)$${\mathrm{Fe}}_{3}O_{4}\left(s\right)\to 3\mathrm{FeO}\left(s\right)+\frac{1}{2}O_{2}\left(g\right)$$

It is also known that FeO acts as a flux to lower the internal melting point [11]. The reduction temperature of Fe_2_O_3_ depends on the oxygen partial pressure according to Liao et al. [23], who found that when the oxygen partial pressure is normal, the Fe_2_O_3_ reduction temperature is 1400 °C. However, when the oxygen partial pressure is low, it is known that the reduction reaction of Fe_2_O_3_ is lowered to 1000 °C. When the oxygen partial pressure is high, Fe_2_O_3_ reduction occurs at 1400 °C, so the reduction of Fe_2_O_3_ does not occur in the sintering temperature (1100–1200 °C), and Fe is present in the form of Fe_2_O_3_. Therefore, the black core bloating mechanism is not activated, and Fe_2_O_3_ and carbon increase the firing temperature and interfere with bloating. When firing under rapid sintering conditions, the surface is quickly sealed due to the liquid phase and carbon is burnt inside the sealed aggregate, with a low oxygen partial pressure inside. Therefore, even if the aggregate bloated by the rapid sintering method is sintered at a high oxygen partial pressure, a black core is observed inside. Finally, FeO forms because the black core formation reaction lowers the melting point of the core portion and promotes viscous behavior, with the pores intensively distributed in the interior.

Figure 5 shows the changes in bulk density and water absorption ratio of acid clay samples fired by the rapid sintering method. In Figure 5, the bulk density starts to decrease at 1150 °C and the water absorption ratio gradually decreases, reaching 1.0% at 1175 °C. In a study by Kang et al. [12], the same raw materials were compared with aggregates bloated by normal sintering and by the rapid sintering method. A black core was generated with the aggregate fired by the rapid sintering method, and at similar densities, the aggregates produced by the black core showed higher water absorption ratios than those not produced in this manner. It was found (Figure 6) that the acid clay has a relatively uniform pore distribution over all temperature ranges, and no black cores were observed. As a result of the observation of the changes in the bulk density and water absorption ratio and the sections of the aggregate, the bloating mechanism of the acid clay is found not to be related to the black core bloating mechanism. Moreover, it can be assumed that the gas contributing to the bloating of the acid clay is created from the detachment of crystalline water from montmorillonite.

### 3.3. Normal Sintering

#### 3.3.1. Bulk Density and Water Absorption Ratio

Lee et al. [24] reported that porous ceramic tiles undergo liquid phase sintering at an elevated temperature and that the sintering rate can be determined by the following equation:
(3)$$-\frac{d\varepsilon }{\varepsilon dt}=\frac{3}{4\eta s}\left(P_{c}-P_{g}\right)$$
(4)$$P_{c}=\frac{-2\gamma }{r}$$(*ε*: porosity, *ηs*: effective viscosity, *P_g_*: gas pressure of closed pore, *P_c_*: capillary pressure of liquid phase, *γ*: surface tension, *r*: diameter of capillary).

According to Equation (3) above, it is advantageous that the capillary pressure of the liquid phase (*P_c_*) is lowered and the internal gas pressure (*P_g_*) is increased for the purpose of increasing the porosity of the aggregate. According to Equation (4), *P_c_* is proportional to the surface tension *γ*, and the surface tension is proportional to the viscous behavior. As the temperature increases and the viscosity of the liquid phase decreases, the capillary pressure of the liquid phase decreases. In other words, in order to improve the porosity of the aggregate, the following points are important:

(1) Due to liquid phase sintering, closed pores are formed in the aggregate.

(2) Gas is generated inside the aggregate.

(3) Appropriate viscosity behavior of the aggregate suitable for expansion should also occur simultaneously.

The pressure inside the aggregate is formed in accordance with the closure of the aggregate surface. The degree of sealing of the aggregate is affected by the green density and the sintering temperature. It is sintered because it cannot be sufficiently sealed at a low temperature below the temperature at which bloating starts and pressure cannot be formed therein. When the surface of the aggregate is densified, gas pressure is generated inside the aggregate. The velocity of the gas generation process is determined by the composition of the raw material. The results of a TG-DTA analysis (Figure 3) show that the acid clay used in this experiment undergoes a weight loss at a high temperature. This is the crystalline water detachment phenomenon of montmorillonite. The detachment of crystalline water at the temperature at which bloating starts increases the pressure inside the aggregate, bloating it. If the detachment of crystalline water occurs slowly, the reaction cannot be completed until the process reaches the temperature at which bloating starts, implying that it can be bloated even at a low heating rate. In this experiment, the aggregate bloated in the sintering condition. This occurred due to the slow detachment of the crystalline water of montmorillonite.

Figure 7 shows the changes in the bulk density and water absorption ratio of the aggregate depending on the temperature ramping time.

In section A, it was observed that the bulk density increased with an increase in the temperature ramping time. In this region, the aggregate is sintered and the measured water absorption ratio is found to be 1% or less indicating that the aggregates undergo viscous behaviors due to liquid phase sintering. According to Riley [8], most types of clay undergo bloating when using a rapid sintering condition. It can be interpreted that the shorter the temperature ramping time is, the more favorable it is for bloating. However, in this experiment, bloating did not occur at a short temperature ramping time; for the bloating of the aggregate, the generation of a foaming gas and proper viscosity behavior must arise in parallel. However, in this area, it is judged that the aggregate does not bloat because viscous behavior cannot sufficiently expand the aggregate, the aggregate surface is not completely closed, and gas pressure (*P_g_*) does not form inside.

In section B, the bulk density decreases as the temperature ramping time becomes longer. When the input temperature is 300 °C, it is observed that the bulk density is slowly lowered in the early stage. It is considered that the bulk density is not abruptly lowered because the viscous behavior of the aggregate is insufficient, as in section A. However, this differs from section A in that the bulk density decreases slowly over time. After a certain period of time, the bulk density of the aggregate decreased rapidly. The same phenomenon was found to occur when the temperature ramping time was 120 min at an input temperature of 600 °C. At both input temperatures, there was a point at which the bulk density was abruptly lowered in section B, which can be explained by the viscous behavior of the aggregate and the generation of bloating gas. According to Kaz’mina et al. [25], a viscosity of 10^7^ dPa·sec or less is required for foaming glass. Moreover, no bloating will occur unless an appropriate viscosity is reached. Under this experimental condition, the longer the sintering time of the aggregate, the more viscous behavior is obtained, with the viscous behavior (*P_c_*) finally converging at 1200 °C. In addition, the longer the temperature ramping time, the less gas generation there will be. Therefore, the optimum bloating-activation zone appears at the point where the viscous behavior (*P_c_*) approaches the maximum value and the gas pressure (*P_g_*) is the highest.

In section C, it was observed that the bulk density of the aggregate was slowly lowered. In the part of section C where the temperature ramping time was long, the aggregate showed appropriate viscous behavior for bloating. However, in order to bloat the lightweight aggregate, it is judged that the gas is insufficient at the temperature at which bloating starts. Acid clay is a material which undergoes slow detachment of crystalline water, but the amount of internal gas generated is reduced as the processing time becomes longer at the calcination temperature. Therefore, gas generation inside the aggregate decreases as the temperature ramping time becomes longer. In this region, the viscous behavior (*P_c_*) is high but the internal pressure (*P_g_*) is low; hence, the bulk density does not drop sharply. However, this leads to over-sintering and the density is gradually lessened.

Both input temperatures are divided into three regions: (A) the sintering zone, (B) the bloating-activation zone, and (C) the over-sintered zone. Each zone is distinguished by the viscous behavior of the body and the difference in the gas pressure inside. The viscous behavior of the aggregate requires time to reach the target viscosity. Therefore, the longer the time, the more advantageous it is. However, because the internal gas pressure decreases as the calcination time increases, the shorter the temperature ramping time is, the more advantageous it is. Because the two conditions are in conflict with each other, regardless of the start temperature, both conditions are satisfied where bloating is activated and section B represents the zone where bloating is activated.

#### 3.3.2. Characteristics of the Pores

The cross-section of the sintered aggregate is shown in Figure 8. It was found that the pores in the aggregate increased gradually as the temperature ramping time was increased. Large pores developed at the input temperature of 600 °C with a sintering time of 180 min, faster than that at 300 °C. The pores were observed to have a relatively homogeneous size.

According to Kose et al. [26], the longer the sintering time is, the larger the internal pores become. According to the classic Ostwald ripening process [27], when pores in a body exist at various sizes, larger pores grow and smaller pores disappear. At this time, the large pores grow rapidly to twice the average pore size, after which the growth rate slows down considerably. As a result, very large pores do not occur and a relatively uniform size is obtained. The aggregates observed in this experiment exhibit a uniform distribution of pores, as the pore size is homogenized by Ostwald ripening before the bloating of the aggregate is caused by the gas generation. When the firing time is prolonged, Ostwald ripening causes the pore growth inside the body to continue, but the speed is very slow, and lengthening the process time is not feasible because doing so incurs a high energetic cost. Therefore, the process should be performed in the bloating-activation zone.

In order to observe the microporous structure of the expanded aggregate in sections B and C, the samples were prepared at an input temperature of 300 °C and at sintering times of 210, 240 and 270 min. Each sample was observed by scanning electron microscope (SEM), and these results are shown in Figure 9. Additionally, mercury intrusion porosimetry (MIP) measurements were taken, as shown in Figure 10. The densities are the lowest at a sintering time of 210 min and are relatively high in the samples for which the temperature was kept high for 240 and 270 min. The observed results show that relatively large pores arise in the image taken at ×100 of the sample held for 210 min. This means that the relatively high gas pressure and viscous behavior in the bloating-activation zone (210 min) simultaneously result in a lower bulk density of the aggregate. The remaining samples showed a significant difference between large pores and small pores, likely due to the relatively low gas pressure and longer time of viscous behavior than in the 210 min sample. For the sample at 210 min in the ×10,000 image, the size of the micro pores was very small, and the sample at 270 min showed larger micropores. This occurred because the longer the sintering time of the aggregate, the more pore growth is caused by Ostwald ripening. The MIP measurement results showed that pores 0.1 to 10 µm in size were not detected in the 210 min and 240 min samples, whereas in the 270 min sample, pores were detected and were actually observed in this section. This is also the case for the 210 min sample, and it expanded by the gas pressure, which was sufficient to produce viscous behavior. For the 270 min sample, expansion arose due to pore growth by Ostwald ripening and over-sintering.

#### 3.3.3. Bloating Index and Single Aggregate Crushing Strength

Samples representing each section of the aggregate at an input temperature of 300 °C were taken; each section is referenced in Figure 7a. These samples are denoted as A (300-60), B (300-210), and C (300-240). The BI index is shown in Figure 11. BI, which is the expansion rate of the aggregate, can determine the sintering and bloating of the aggregate because it shows the corresponding shrinkage and expansion. As a result, the specimen of section A contracted more than the shaped body. This indicates that the aggregates are sintered in the A region. B and C expanded somewhat and the B sample shows somewhat greater expansion than C. This shows that the BI results tend to be similar to the density measurement results. As shown in Section 3.3.1, zone B is advantageous for the expansion of the aggregate.

The results of the single aggregate crushing strength measurement of the aggregate are shown in Figure 12. The single aggregate crushing strength of the aggregate was highest in the A (300-60) samples, followed by C (300-240) and B (300-210). The A (300-60) samples showed the highest single aggregate crushing strength because they had low porosity. The B (300-210) and C (300-240) samples with relatively low densities were also low in strength. However, the experimental results of Gonzalez-Corrochano et al. [18] show that the B (300-210) sample is similar for use as a structural lightweight aggregate material because the single aggregate crushing strength of a commercially available aggregate is only 3 MPa. The single aggregate crushing strength of the aggregate tends to decrease as the density decreases.

## 4. Conclusions

In this study, to clarify the bloating mechanism under normal sintering conditions of a ceramic body and to search for appropriate bloating conditions, aggregates were sintered by normal sintering and rapid sintering methods using acid clay. The bulk density, bloating index, water absorption ratio and crushing strength were then measured and the microstructure was observed. The single aggregate crushing strength was measured. In this study, the following conclusions were determined.
(1)The bloating mechanism of the acid clay is not related to the typical reactions of lightweight aggregates in which a black core is formed. It is a combination of Ostwald ripening and detachment of the crystalline water of montmorillonite minerals. Detachment of crystalline water of montmorillonite is continuous at high temperature.(2)When the sintering rate is very rapid and the supplied amount of energy is low, the aggregate does not show sufficient viscous behavior and bloating does not occur. (A: Sintering zone.)(3)In order to bloat the aggregate, sufficient viscous behavior and gas generation should be properly combined, so there is an optimal bloating-activation zone because the two conditions are in conflict with each other. (B: Bloating-activation zone.)(4)When the sintering rate is low, the viscous behavior for bloating is sufficient. However, bloating is not adequate because the calcination section becomes longer and the internal pressure is lowered. The aggregate is over-sintered in this section. (C: Over-sintering zone.)

Given that the aggregate cannot be manufactured over extended times at actual sites, it is necessary to carry out the process in the bloating-activation zone.

## Figures and Tables

**Figure 1 materials-12-00267-f001:**
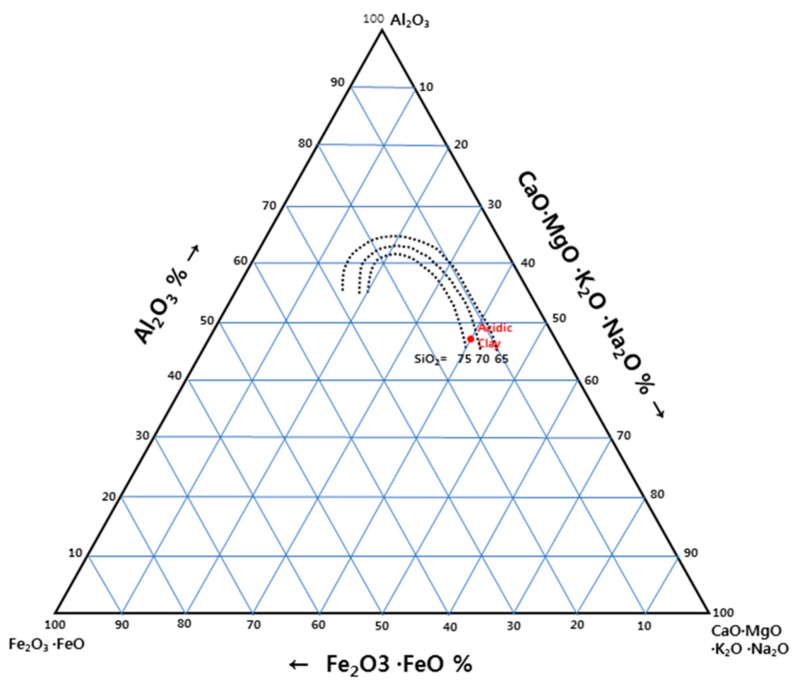
Cougny diagram of acid clay [9].

**Figure 2 materials-12-00267-f002:**
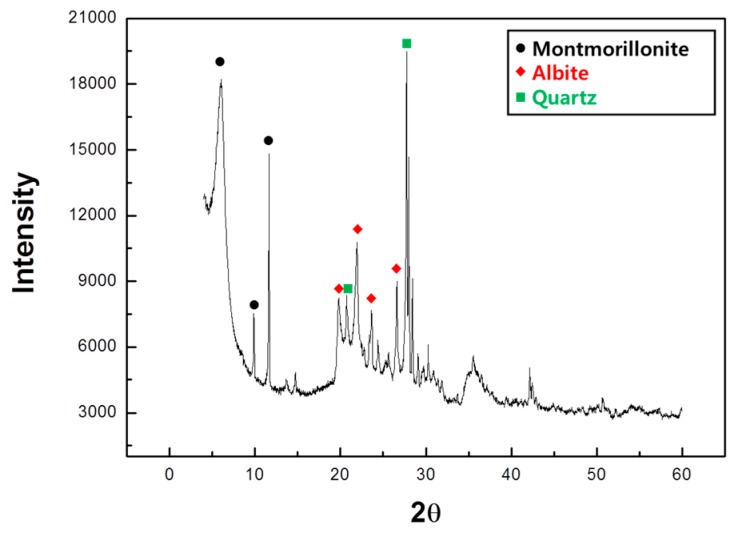
X-ray powder diffraction (XRD) results of for acid clay.

**Figure 3 materials-12-00267-f003:**
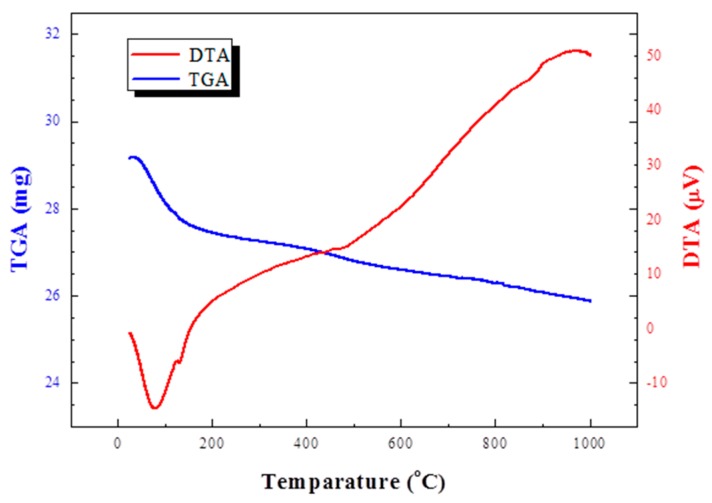
Thermogravimetry-differential thermal Analysis (TG-DTA) results of acid clay.

**Figure 4 materials-12-00267-f004:**
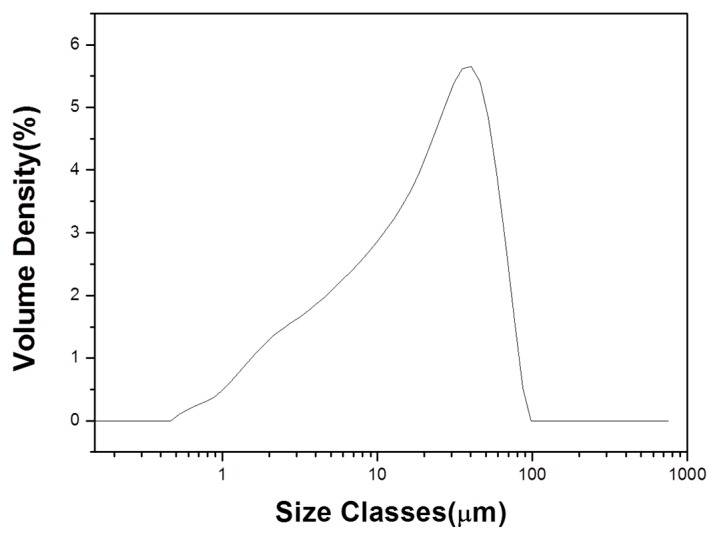
Particle size analysis results of raw acid clay.

**Figure 5 materials-12-00267-f005:**
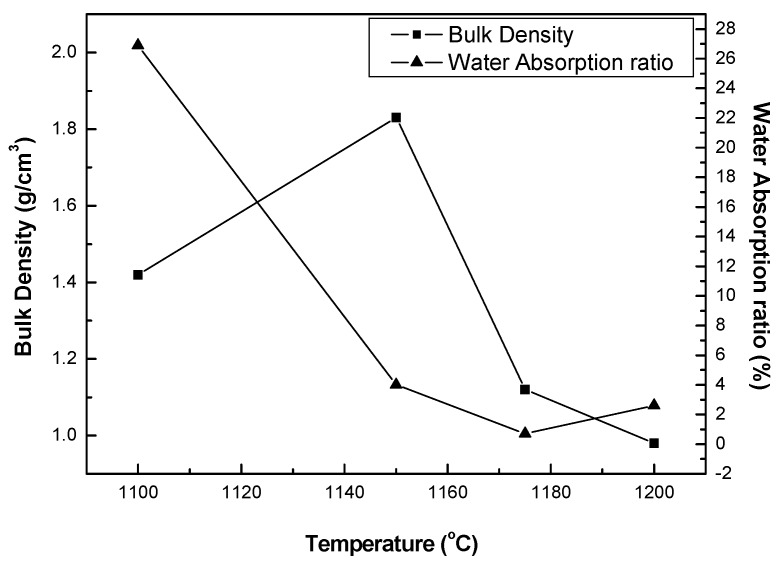
Bulk density and water absorption ratio of the sample fired by the rapid sintering method.

**Figure 6 materials-12-00267-f006:**
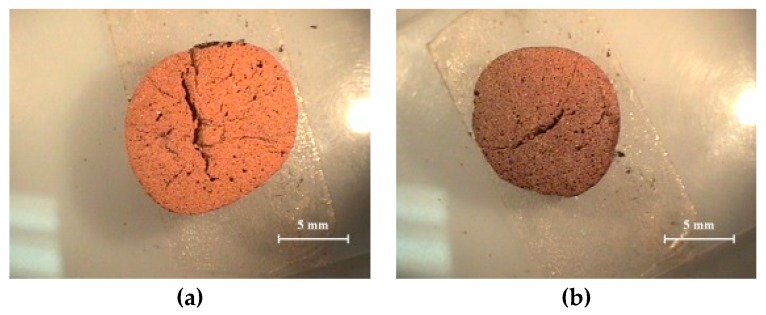
Cross section of the aggregate fired by the rapid sintering method: (**a**) 1100 °C; (**b**) 1150 °C; (**c**) 1175 °C; (**d**) 1200 °C.

**Figure 7 materials-12-00267-f007:**
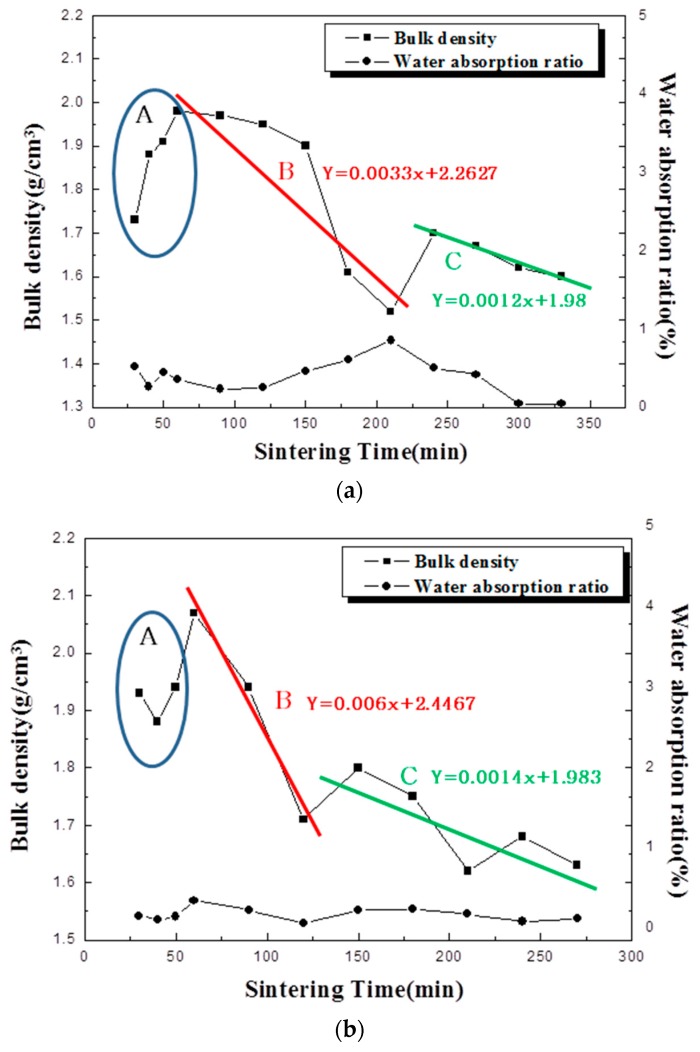
Bulk density and water absorption change of the aggregate according to sintering time: (**a**) input temperature 300 °C and (**b**) input temperature 600 °C.

**Figure 8 materials-12-00267-f008:**
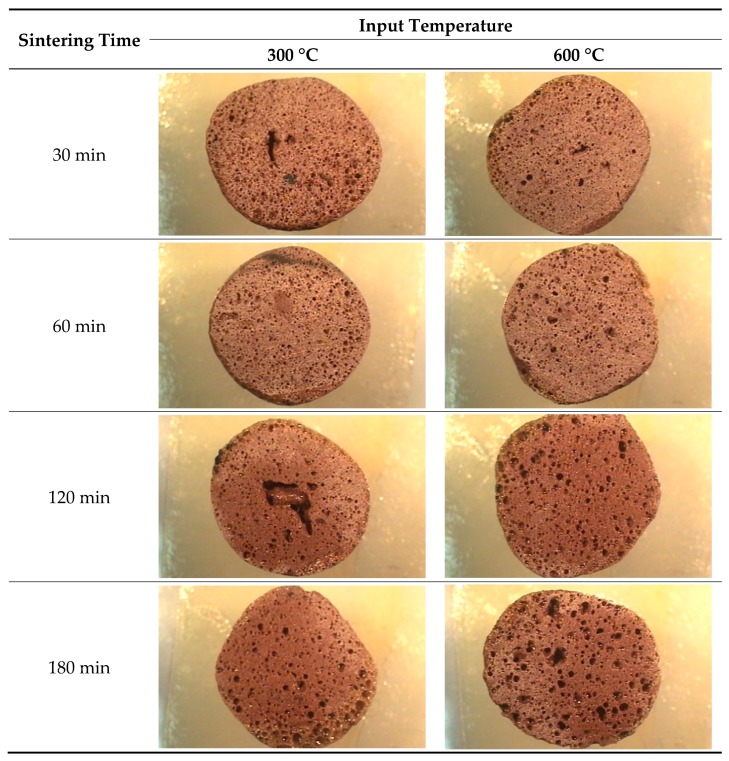
Cross-section of the aggregate according to the heating time.

**Figure 9 materials-12-00267-f009:**
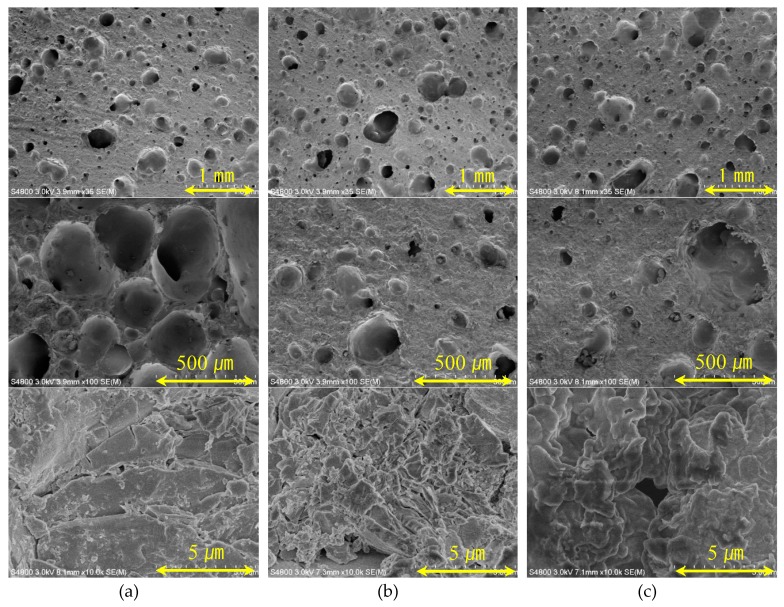
Scanning electron microscope (SEM) observation results of samples at firing input temperature 300 °C: (**a**) 210 min, (**b**) 240 min, and (**c**) 270 min.

**Figure 10 materials-12-00267-f010:**
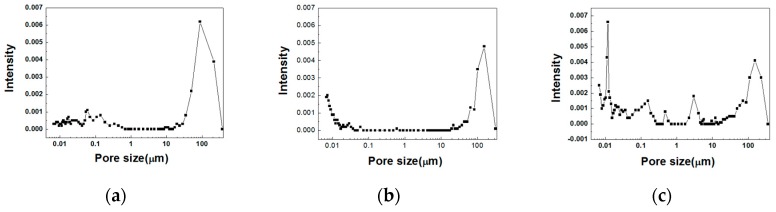
Mercury intrusion porosimetry (MIP) measurements of samples at a firing start temperature of 300 °C: (**a**) 210 min, (**b**) 240 min, and (**c**) 270 min.

**Figure 11 materials-12-00267-f011:**
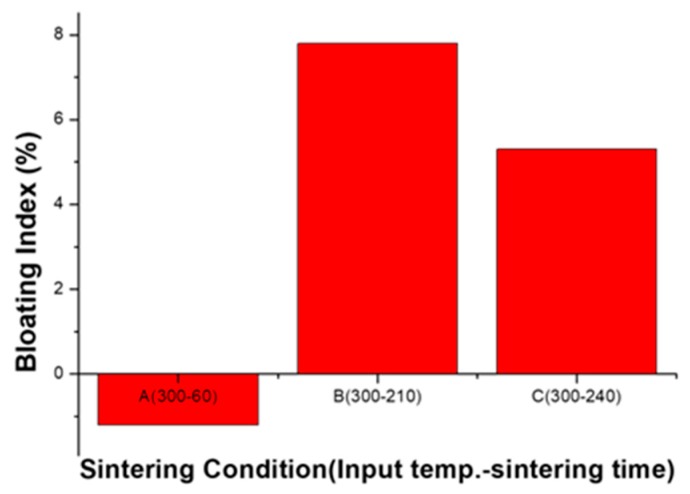
Bloating Index (BI) of the sintering aggregate.

**Figure 12 materials-12-00267-f012:**
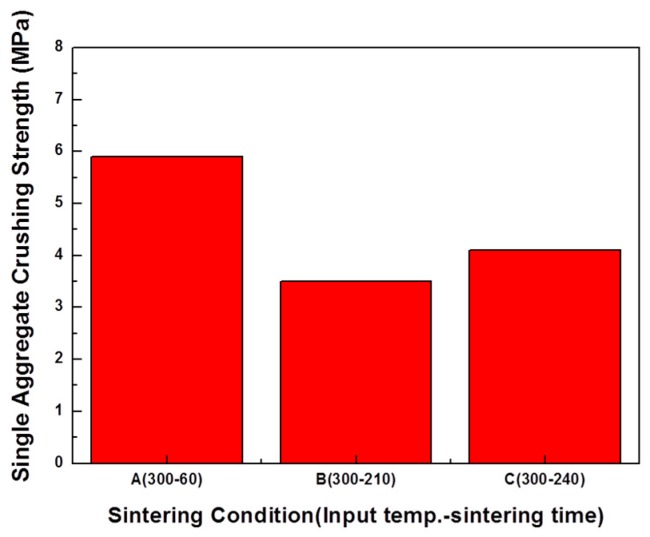
Single aggregate crushing strength test result.

**Table 1 materials-12-00267-t001:** Compositions of the acid clay (wt %).

Items	Acid Clay
SiO_2_	67.3
Al_2_O_3_	12.9
Fe_2_O_3_	2.9
CaO	2.8
MgO	2.4
Na_2_O	1.6
K_2_O	1.8
TiO_2_	0.6
P_2_O_5_	0
Ig-Loss	7.7
Total	100

**Table 2 materials-12-00267-t002:** Sintering conditions for artificial lightweight aggregates.

Input Temperature (°C)	Maximum Temperature (°C)	Sintering Time (min)
300	1200	30, 40, 50, 60, 90, 120, 150, 180, 210, 240, 270, 300, 330
600		30, 40, 50, 60, 90, 120, 150, 180, 210, 240, 270

**Table 3 materials-12-00267-t003:** Liquid limit (LL), plastic limit (PL) and plasticity index (PI) of acid clay.

Liquid Limit (LL)	Plastic Limit (PL)	Plasticity Index (PI)
52	38	14

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
