# Peer review of "Optimum Bloating-Activation Zone of Artificial Lightweight Aggregate by Dynamic Parameters"

_materials, 2019, doi:10.3390/ma12020267_

Round 1
Reviewer 1 Report
1. XRD is confusing. The patterns should be provided from at least 3 degrees in order to verify clay minerals presence. Montmorillonite’s and mica’s presence are not proven. In any case smectite’s presence cannot be proven only by XRD of the bulk sample.
2. XRD: The peak at about 55 degrees is not a clay minerals peak (it is too sarp)
3. XRD: From Fig. 2 it seems quartz to be present while other impurities are also present.
4. The presence of smectite is not proven while the authors discuss that smectite is montmorillonite. What techniques the authors use to identify montmorillonite?
5. The authors should improve English e.g. line 85.
6. The authors should correct grammar and format errors e.g. lins 85, 99, 116, 256.
7. The authors should add some references e.g. line 90.
Author Response
1,2,3,4 As you pointed out, XRD alone can not detect montmorillonite. However, we have determined that the Montmorillonite phase exists because of the information provided by the supplier and the results of the TG-DTA analysis.
http://www.donghaech.com/product/product01.htm
This is an excerpt from the upper link.
(Acid clay based on montmorillonite (Ca-Montmorillonite) is chemically reacted to improve adsorption ability, decolorization ability and catalytic ability, and is used for edible oil, refining and adsorption of petrochemical products, It is a product of minerals.
5,6,7 : The thesis was commissioned by a specialized company and the whole revision was made.
Reviewer 2 Report
This paper presents an experimental investigation to compare the bloating mechanism of artificial lightweight aggregate under sintering and direct firing conditions. The following comments are suggested:
1. The authors should cite more recent publications in their literature review.
2. In the literature review, it should be emphasized that the sintering conditions mainly include preheating temperature, preheating time, sintering temperature, sintering time and cooling rate, which are closely related to the sintered aggregate.
3. In addition to the chemical composition, the physical properties (such as density or specific gravity, water absorption) of the acidic clay used in this experiment should be provided.
4. The resolution of Figure 1 is not good and should be improved.
5. In Equation 2, each parameter should be defined.
6. In lines 153-155: “V. A. Lotov [15] and others have suggested that the body must reach a suitable viscosity for bloating, if the viscosity is not reached, it is said that bloating is not possible”. In the sentence, a suitable viscosity should be quantified to express its range. Please correct this sentence.
7. In the article, the meaning of t1, t2, and t3 in Figure 6 should be explained.
8. Except bulk density and water absorption ratio, the bloating index and the crushing strength of the sintered aggregate should be considered.
9. The bulk density of the sintered aggregate does not meet the requirements of the ASTM C330 (the bulk density of lightweight coarse aggregates is less than 880 kg/m3). Authors should give more explanations for this.
Author Response
1. I cited an additional paper from a recent study.[7],[8]
2. In the introductory section, the chemical properties of the bloat (line 31-38), and the process aspects (lines 39-48) were described separately and described the importance of the sintering method.(line 53-59)
3. The PSA (Particle Size Analysis) measurement result of the raw material is shown(Fig.4), and the Atterburg limit is measured and shown. (Table 3)
4. I redrawn the picture again. (fig.1)
5. All parameters are defined and shown on lines 185-186.
6. References are cited and replaced, This is described on line 230-232.
7. It is indicated in fig.9.
8. The expansion index (BI) (fig.12) and fracture strength (fig.13) were measured and shown. And described in line 309-325.
9. The density you have presented is different from the results presented in this paper and the definition of density is detailed on lines 89-90. Lightweight aggregate density includes loose bulk density and particle density. In ASTMC330, loose bulk density is defined as 880kg / m3 or less. In EN13055, loose bulk density is less than 1200kg / m3 and particle density is defined as 2000kg / m3 or less.
Reviewer 3 Report
The topic is interesting, but, in my opinion, the article should not be published in its current form. English is very poor in many sections of the manuscript, which makes it difficult to understand the manuscript content. Similarly, the narrative line referring to the construction of the sentences is quite deficient and there are times when punctuation marks have been used incorrectly (for example, line 16: "...longer, When..." when it should be "....longer. When..."; another example on line 56). This makes even revision very difficult, so my recommendation is that authors review and rewrite the entire article with the help of a native English speaker.
In addition, I recommend revising and changing certain terminology; for example, the term "acidic clay". This term sounds rather strange. Is this clay really different from other clays used in the manufacture of LWAs? If there really isn't any difference, maybe it's best to simply define it as "clay", removing the term “acidic”.
The authors also talk about "direct firing". What exactly do they mean and what is the difference with respect to "sintering"? If there is no difference, please eliminate such a term.
Please, although the introduction is not bad, I suggest that it is improved with the inclusion of more current works on the production of lightweight aggregates, such as those by Moreno-Maroto et al. (2017, 2018) or Ayati et al. (2018).
The authors place great emphasis on the role of Fe2O3 in bloating, however, there are other factors and components that also favor it and have been ignored. Look for information and improve the parts of the manuscript that refer to this aspect.
Section 2.1 is very poor, and is not deepened for example in the conditions used in either the XRD or the TG-DTA test. Similarly, a basic characterization of the clay should be carried out, especially of its particle size distribution and its Atterberg limits. Please, provide all this information.
Section 2.2. Explain the reason for using a diameter of 10 mm (include bibliographic citation if necessary). Explain which pelletizing method was used. Was the sample extruded? Were the pellets shaped by hand or using a pelletizer? Please specify.
Lines 70-75: These lines are quite confusing and the cooking procedure needs to be better explained. How long were the pellets kept in the 300 or 600 ºC zone? Also specify why you used a temperature of 1200 ºC.
Line 74: What heating ramp did you use for direct firing? (However, I think direct firing is not a correct term, if you used a non-rotating oven, it may be best to indicate that the firing was done under static conditions).
Please do not write Montmorillonite in capital letters in some lines (e.g. line 132) if other lines show it lowercase: montmorillonite (e.g. line 127). Always apply the same narrative criterion.
Improve Figure 4, indicating what A, B and C mean in the figure caption. Is it appropriate to use the term "temperature rise time" on the x-axis? Wouldn't it be better to simply indicate "sintering time"?
Lines 138-139: Just to clarify, since these lines seem confusing. Short firing times are used when the temperature used is very close to the melting of the material. These high temperatures favor that the viscosity is adequate for the retention of the generated gases, however, it cannot be prolonged in the time since otherwise the process would not be operative, as the aggregates can stick to others or to the tube of the kiln. In your case, and taking into account the long sintering times you have used, the temperature does not seem to be operatively at the "limit", so it is normal that at short firing times not enough liquid phase has yet formed. Perhaps a more complete study would have covered the use of different firing temperatures in the rotary kiln.
Please include a new section after section 2.2, which should be section 2.3 in which you explain the methods you have followed in the characterization of the sintered aggregates. It would be good if you included more tests besides bulk density or porosity, because to be honest, the characterization made in the study is really poor (it has not been analyzed even something as basic in the LWAs as is the bloating index).
Line 196: MIP. Please enter the full name the first time the abbreviation is used.
Please improve the conclusions according to the changes implemented in the rest of the manuscript.
Author Response
1.This paper was rivised by professional company.
2. http://www.donghaech.com/product/product01.htm
This is an excerpt from the upper link.
(Acid clay based on montmorillonite (Ca-Montmorillonite) is chemically reacted to improve adsorption ability, decolorization ability and catalytic ability, and is used for edible oil, refining and adsorption of petrochemical products, It is a product of minerals.
3. The term has been modified. (direct firing → rapid sintering) Two different sintering methods were used and the contents are shown on line 80-86.
4. Your recommended papers have been added to the reference [7], [8]
5. In the introductory section, the chemical properties of the bloat (line 31-38), and the process aspects (lines 39-48) were described separately and described the importance of the sintering method.(line 53-59)
Section 3.2, lines 161-173 of the paper, Indicated that the foaming mechanism of acid clay is not the formation of black cores.
6. The PSA (Particle Size Analysis) measurement result of the raw material is shown(Fig.4), and the Atterburg limit is measured and shown. (Table 3)
7. The aggregate is handmade and is described on line 78.
The size of the aggregate was molded to 10 mm, which is not too small by reference [11].
8. Sintering method is expressed in line 83-86 of the paper.
Sintering was carried out. In order to simulate the firing conditions of a rotary kiln in the sintering experiment, the input temperatures were fixed at 300℃ and 600℃, the temperature was raised to 1,200℃ immediately and firing was then done but not then maintained at a maximum temperature.
9. A box type electric furnace was used and was represented on line 80.
10. It has been modified.
11. It has been modified.
12. I agree that the aggregate will be close to the melt during bloating. And we are working on a solution to the problem. In this study, we studied the conditions under which bloating occurs. Melting are not covered in this paper.
13. The expansion index (BI) (fig.12) and fracture strength (fig.13) were measured and shown. And described in line 309-325.
14. MIP (mercury intrusion porosimetry) is shown on line 284.
15. The conclusions were revised.
Round 2
Reviewer 1 Report
I believe that the authors answer my minor comments (5-7) but did not answer any of my major comments (1-4), so I repeat them:
1. XRD is confusing. The patterns should be provided from at least 3 degrees in order to verify clay minerals presence. Montmorillonite’s and mica’s presence are not proven. In any case smectite’s presence cannot be proven only by XRD of the bulk sample.
2. XRD: The peak at about 55 degrees is not a clay minerals peak (it is too sarp)
3. XRD: From Fig. 2 it seems quartz to be present while other impurities are also present.
4. The presence of smectite is not proven while the authors discuss that smectite is montmorillonite. What techniques the authors use to identify montmorillonite?
Author Response
1,2,3,4 I repeat the XRD measurement and acknowledge the part you said. For Montmorillonite, a low angle peak should be observed. Montmorillonite peak was detected near 2 ° after re-measurement. And albite and quartz were also detected. I have taken a look at it up to line 108-111. Montmorillonite was determined based on the XRD peak, the data provided by the supplier, and the TG-DTA results.
Reviewer 3 Report
Firstly, I would like to congratulate the authors, as I believe that the study addresses certain very interesting aspects in the production of lightweight aggregates (LWAs). The manuscript has been improved compared with the original version. However, I have still found quite a few areas for improvement. Please read my comments carefully, as in many of them, I try to help improve the writing of the article and especially some key points, such as those related to expansion processes. I think that the article could be improved according to my contributions. In any case, as I say, the work is correct and of interest to LWAs research.
Line 26: …shale. Recently,…
Line 28: [6] or carbon fiber…
Line 29: For lightweight….
Line 29: …processes, Riley [8]…
Line 40: agglomerate? Are the authors referring to “pellet” or “granule”? If so, change it by one of the terms proposed.
Line 61: …materials. In order to…
Lines 70-71: Remove “analysis” at the end of the phrase as follows: A XRD (Miniflexll, Rigaku, Japan) analysis was performed to assess the crystal phase. After…
Line 73: Please, specify if the mastersizer 3000E device determines the particle size distribution by laser diffraction or other mechanism.
Line 78: were shaped by hand
Lines 83-86: The sintering procedure using the rotary kiln is still unclearly explained. Several points to be clarified:
- Why do the authors begin with “In the second step, sintering was carried out”. It is important to note that “sintering” was also conducted when using the electric furnace according to the first protocol indicated in the previous lines.
- What kind of rotary kiln was employed?
- What do the authors refer to when they say in line 86 that “but not then maintained at a maximum temperature”? So, how long the pellets are exposed at the maximum temperature (1200ºC)? What is the temperature applied after the maximum? These lines are very confusing.
Line 89: [15], where
Line 90: [16]. The cross…
Line 95: Replace the term “compressive strength” by “crushing strength” or “single aggregate crushing strength”. Change it in the rest of the manuscript when applicable.
Line 95: What device (trademark, manufacturer) did you use to perform the crushing test?
Line 101: of the raw acid clay
Line 105: due to another
Line 120: It is not Cougney, but Cougny and write the reference number right next to it.
Line 120-121: It is worth remembering that Cougny study does not indicate the protocol followed to determine the most appropriate particle size distribution (PSD). According to the diagram developed by Cougny on PSD, it is highly likely that the protocol he followed was that related to sedimentation tests, which tend to give rise to much finer PSD than the methods based on laser diffraction. Therefore, Cougny criterion on PSD is only tentative.
Line 121: …aggregates. Nevertheless, the analysis showed…
Line 122: microparticles instead of nanoparticles
Line 123-125: However, the results of Fig. 4 shows that the PSD is relatively coarse for a clay. According to Fig. 2, it could be related to the mica content of the sample. These macro-particles of mica could have affected negatively to the plasticity, because although the authors say that the plasticity was adequate for pelletizing, the ratio PI/LL is relatively low if compared to those of clays (PI/LL is usually higher than 0.33 and mainly to 0.5 in clayey materials and clays respectively according to the paper “What is clay? A new definition of “clay” based on plasticity and its impact on the most widespread soil classification systems” of Moreno-Maroto and Alonso-Azcárate, 2018). Probably the small amount of montmorillonite has acted as a binder helping to improve the workability of the material, but obviously the plasticity is significantly lower than a that of a pure montmorillonite. I encourage the authors to think about it in order to improve this part of the manuscript.
Line 139: The results of LL and PL are wrong. LL is always higher than PL. I suppose that LL is actually 52 and PL is 38. Please, revise it.
Lines 145-149: Include the reference number from which the iron reduction reactions have been extracted. On another note, the formation of CO from the combustion of organic matter under rather reductive conditions could act as a reducing agent to form FeO. The paper of Ki Gang Lee (one of the authors of this manuscript) explains this phenomenon very well in its Introduction: Lee, 2016: Bloating Mechanism of Lightweight Aggregate with the Size.
Line 150: “It is also known that FeO acts as a flux to lower the internal melting point”. Please include reference for this statement.
Lines 154-155: “Fe2O3 reduction occurs at 1,400℃, the reduction of Fe2O3 does not occur”. These two sentences are contradictory. Does iron reduce or not? Remove the sentence that is wrong, please.
Lines 153-162: Indicate reference/s and improve the narrative line of these line, which are somewhat cumbersome in their current form.
Line 164: remove: “, and the cross section is shown in Fig.6.”
Lines 164-165: …starts to decrease at…
Line 165: “reaching nearly 0”. What is 0 here? 0 %? Improve the sentence.
Lines 173-175: the authors should indicate that this effect would concur with the mechanism of dehydration and dehydroxylation of the montmorillonite, according to the DTA-TG curve of Fig. 3, including the necessary reference/s.
Line 219: most types of clay undergo
Line 234: According to Kaz'mina et al. Please include a reference number.
Line 235-236: Do the authors know what is the approximate gas pressure (Pg) range required for a LWA to bloat? If so, please include the data and reference if applicable. The same for the capillary pressure of liquid phase (Pc).
Line 245: VERY IMPORTANT: It can be true that bound water (hygroscopic water can be discarded) can play an important role in the formation of pores, but the reviewer thinks that it is more likely that an important proportion of the pores can be formed because of the release of water molecules from the dehydroxylation of the clay minerals. Dehydration usually takes place at temperatures well below those of dehydroxylation, so probably the water from dehydration has been lost before reaching the point at which the material reaches the proper viscosity to trap the gas. My recommendation is that the authors should review the bibliography on DSC-TG or similar thermal tests in montmorillonite and mica, in order to improve the discussion and conclusions of the paper. Similarly, perhaps mica is also important, since the authors have focused only on montmorillonite and they have forgotten that mica has a particle size (usually macroscopic) that is significantly larger than that of montmorillonite (usually < 1 microns). It means that the “destruction” of the mica particle could need more energy in proportional terms than that of montmorillonite, i.e., mica particles could be decomposed more slowly due to their larger particle size. Furthermore, the XRD results (Fig. 2), the PSD (Fig. 4) and the relatively low plasticity point out that the representation of mica is more significant than that of montmorillonite. Please, think about it, look for information in bibliography and complete the explanation on pore formation, that is the main point of the paper.
Line 264: and section B represents the zone where bloating is activated.
Line 271: Include the reference number of Klose et al.
Line 274-276: “At this time, the large pores grow rapidly to twice the average pore size”: this sentence means that large pores are formed. “As a result, very large pores do not occur and a relatively uniform size is obtained”: by contrast, this sentence means that large pores are not formed. Please, clarify the concepts.
Line 280: incurs a high cost. What kind of cost do the authors refer? Economic, energetic, environmental…? Please, specify.
Line 289: “are the lowest” or “are the lower” depending on what the authors want to mean. The term “are lowest” is not adequate.
Line 306 (Fig. 9 caption): Diffusion of pressure into larger pores of small pores? This sentence does not make sense.
Line 314: Samples representing
Lines 318-319: This indicates that the aggregate is sintered in this region…which region do the authors refer to here? Please, specify if it refers to region A according to Fig. 7. You could introduce that the samples A, B and C are called this way because they have been sintered in according to the zones A, B and C of Fig. 7. Introduce it in line 315 after the phrase “These samples are denoted as A (300-60), B (300-210), and C (300-240)”.
Fig. 13, Line 322 and successive lines when applicable: replace the term compressive strength by crushing strength in case you have performed the test through crushing individual granules.
Line 324: “The A (300-60) samples showed the highest compressive strength because they were sintered”. Such statement is wrong. In fact, the aggregates sintered for longer times have undergone greater sintering than the A (sintered only for 60 min). Think about it. It is most likely that A is more strong than B and C because A is less porous than B and C (porosity affects negatively mechanical strength).
Lines 326-329: the narrative line should be improved. In addition, the authors cannot affirm that the aggregate is suitable for structural concrete without performing tests on concrete specimens. The authors should simply indicate that the results of crushing could point out that the aggregate could be suitable in lightweight structural concrete, since the results exceed those found in the bibliography about commercial LWAs. But obviously, this is just a conjecture that should be proven by additional tests in real concrete specimens.
Line 328: MPa instead of Mpa
Line 337: Fig. 13: Y-axis: MPa instead of Mpa
Line 339: of a “body”? Please replace it by another term, for example “ceramic body” or “clay”.
Line 340: aggregates were sintered
Lines 341-342: Replace “The bulk density, bloating index, and water absorption ratio were then measured and the microstructure was observed. The compressive strength was measured” by “The bulk density, bloating index, water absorption ratio and crushing strength were then measured and the microstructure was observed”.
Line 344: Improve the sentence. For example: The bloating mechanism of the acid clay is not related to the typical reactions of lightweight aggregates in which a black core is formed.
Line 345: Take into account the reviewer’s comment on the role of phyllosilicates dihydroxylation including mica minerals as alternative source of gas.
Line 350: …combined, so there is…
Line 353-354: However, bloating is not adequate because the calcination section becomes longer and the internal pressure is lowered.
Author Response
All the simple modifications you requested have been revised.
Below we will address the key issues you have requested.
Lines 83-86: The sintering procedure using the rotary kiln is still unclearly explained. Several points to be clarified:
- Why do the authors begin with “In the second step, sintering was carried out”. It is important to note that “sintering” was also conducted when using the electric furnace according to the first protocol indicated in the previous lines.
- What kind of rotary kiln was employed?
- What do the authors refer to when they say in line 86 that “but not then maintained at a maximum temperature”? So, how long the pellets are exposed at the maximum temperature (1200ºC)? What is the temperature applied after the maximum? These lines are very confusing.
A : The rotary kiln was not used in this paper. The firing methods used in the paper are rapid firing and normal sintering. We have fixed the term to avoid confusion. After Normal sintering, colling is carried out naturally. This is shown on line 87.
Line 120-121: It is worth remembering that Cougny study does not indicate the protocol followed to determine the most appropriate particle size distribution (PSD). According to the diagram developed by Cougny on PSD, it is highly likely that the protocol he followed was that related to sedimentation tests, which tend to give rise to much finer PSD than the methods based on laser diffraction. Therefore, Cougny criterion on PSD is only tentative.
A : I agree with your opinion. Acid clay has a low microparticle ratio but is well formed and foamed. This part has been shown up to 128-130.
Q : Line 150: “It is also known that FeO acts as a flux to lower the internal melting point”. Please include reference for this statement.
A : Modified and displayed on line 156.
Lines 154-155: “Fe2O3 reduction occurs at 1,400℃, the reduction of Fe2O3 does not occur”. These two sentences are contradictory. Does iron reduce or not? Remove the sentence that is wrong, please.
A : The reduction of Fe2O3 takes place at 1400 ℃ in an oxidizing atmosphere and at 1000 ℃ in a reducing atmosphere. Therefore, in the oxidizing atmosphere, the reduction reaction does not occur at the process temperature (1100 ~ 1200 ℃). This is shown on line 156-162.
Line 235-236: Do the authors know what is the approximate gas pressure (Pg) range required for a LWA to bloat? If so, please include the data and reference if applicable. The same for the capillary pressure of liquid phase (Pc).
A : Appropriate Pg and Pc values are not shown.
Line 245: VERY IMPORTANT: It can be true that bound water (hygroscopic water can be discarded) can play an important role in the formation of pores, but the reviewer thinks that it is more likely that an important proportion of the pores can be formed because of the release of water molecules from the dehydroxylation of the clay minerals. Dehydration usually takes place at temperatures well below those of dehydroxylation, so probably the water from dehydration has been lost before reaching the point at which the material reaches the proper viscosity to trap the gas. My recommendation is that the authors should review the bibliography on DSC-TG or similar thermal tests in montmorillonite and mica, in order to improve the discussion and conclusions of the paper. Similarly, perhaps mica is also important, since the authors have focused only on montmorillonite and they have forgotten that mica has a particle size (usually macroscopic) that is significantly larger than that of montmorillonite (usually < 1 microns). It means that the “destruction” of the mica particle could need more energy in proportional terms than that of montmorillonite, i.e., mica particles could be decomposed more slowly due to their larger particle size. Furthermore, the XRD results (Fig. 2), the PSD (Fig. 4) and the relatively low plasticity point out that the representation of mica is more significant than that of montmorillonite. Please, think about it, look for information in bibliography and complete the explanation on pore formation, that is the main point of the paper.
Line 345: Take into account the reviewer’s comment on the role of phyllosilicates dihydroxylation including mica minerals as alternative source of gas.
A : I agree with your thoughts about the impact of pyrophyllite minerals. The dehydration reaction of pyrophyllite-based minerals is the key mechanism for the expansion of acid clay. To make it clearer, I re-measured the XRD. As a result, montmorillonite was observed but no mica was found. So I thought it was inappropriate to discuss the impact of mica. However, in response to your feedback, I have referred to the crystallization dehydration of pyrophyllite-based minerals in line with Riley's study on line 120-123.
Line 274-276: “At this time, the large pores grow rapidly to twice the average pore size”: this sentence means that large pores are formed. “As a result, very large pores do not occur and a relatively uniform size is obtained”: by contrast, this sentence means that large pores are not formed. Please, clarify the concepts.
A : According to the classic Ostwald ripening process [29], when pores in a body exist at various sizes, larger pores grow and smaller pores disappear. At this time, the large pores grow rapidly to twice the average pore size, after which the growth rate slows down considerably. As a result, very large pores do not occur and a relatively uniform size is obtained. (This has been revised in line 277-280.)